# Effects of Rotational Speed on the Microstructure and Mechanical Properties of 2198-T8 Al-Li Alloy Processed by Friction Spot Welding

**DOI:** 10.3390/ma16051807

**Published:** 2023-02-22

**Authors:** Zheng Pang, Jin Yang, Yangchuan Cai

**Affiliations:** 1Jiang Su Bao Steel Fine Wire @ Cord Co., Ltd., Jiangsu 226114, China; 2School of Materials Science and Engineering, Tianjin University of Technology, Tianjin 300384, China

**Keywords:** aluminum-lithium alloy, mechanical properties, friction stir welding, rotational speed, microstructure

## Abstract

The friction spot welding (FSpW) method was used to weld 2198-T8 Al-Li alloy at rotational speeds of 500 rpm, 1000 rpm, and 1800 rpm. It was shown that the grains in the FSpW joints were transformed from “pancake” grains to fine equiaxed grains by the heat input of welding, and the reinforcing phases of S’ and θ were all redissolved into the Al matrix. This leads to a decrease in the tensile strength of the FsPW joint compared to the base material and a change in the fracture mechanism from mixed ductile-brittle fracture to ductile fracture. Finally, the tensile properties of the welded joint depend on the size and morphology of the grains and their dislocation density. At the rotational speed setting of 1000 rpm in this paper, the mechanical properties of welded joints consisting of fine and uniformly distributed equiaxed grains are best. Therefore, a reasonable set of the rotational speed of FSpW can improve the mechanical properties of the welded joints of 2198-T8 Al-Li alloy.

## 1. Introduction

Friction stir welding (FSW) is a solid−state joining technology that was invented in 1991 by the Welding Institute (Britain). It can avoid the defects caused by melting and solidification of the welded alloy effectively, leading to a higher quality of the welded joints [1,2,3]. It is well established in the field of welding light alloys such as Al alloy, Mg alloy, and their alloys [4]. Hence, this technique is also gradually applied to similar high melting points or high strength materials such as Ti6Al4V or NiTi, where good performance has been achieved [5,6,7].

Friction spot welding (FSpW) is developed from friction stir welding. In the FSpW process, the moving part is replaced by the insertion and recovery part of the combined stirring head. [8,9] Unlike conventional friction welding, friction spot welding can be used to weld more types of materials and machined surfaces. It has advantages such as high energy efficiency, high surface quality, good performance, and being environmentally friendly. In addition, FSpW can be used to repair hole defects and cracks inside the weld produced by friction stir welding [10,11,12,13]. Friction spot welding can be divided into two categories, depending on the type of workpiece inserted: the stir needle−insertion type and the shaft shoulder−insertion type. The shaft shoulder−insertion type friction spot welding features a larger weld nugget, and the welding joint is subject to shear stresses of high intensity [14]. Sergio et al. [15] investigated the effects of rotating speed and welding time on the properties of 2024−T3 lap joint. Their research results suggest that the weld nugget is mainly composed of fine dynamic re−crystallization resulting from the large shear strain and high processing temperature during the FSpW process. When the welding time was set to 5.8 s and the rotational speed was set to 2400 rpm, a welding joint with outstanding mechanical properties could be obtained. Shen et al. [16] characterized the microstructure of the welding joint of 7075−T6 aluminum alloy welded via the friction spot welding method. The mechanical properties of the welding joint were also studied. The conclusion shows that coupling lower rotational speed and short welding time can guarantee excellent mechanical properties of the welding joint, prolonging welding time at a higher welding rotational speed; meanwhile, it is beneficial in terms of obtaining good mechanical properties.

Advantaged by their high strength and stiffness, Al−Li alloys offer the potential for substantial weight saving in aero−structural components, since they allow a massive weight reduction. FSpW technology is now considered to be the ideal welding method for Al−Li alloys. It features low heat input, as well as the prevention of the loss of the lithium (Li) element and the defects caused by the solidification of the welding joint. These advantages result from the fact that metal melting does not occur [17,18]. The effect of the three principal friction welding parameters of pin geometry, welding, and rotating speeds on the joint performance was investigated by Ahmadi et al. [19], and it was concluded that higher rates of rotational speed to welding speed lead to coarser and larger grains and confirmed welded samples’ weak or strong performance.

In this study, the microstructure modifications of FSpW−procced 2198−T8 Al−Li alloy were studied. The corresponding hardness and tensile properties of the samples were tested, and the effects of rotation speed (500 rpm, 1000 rpm, and 1800 rpm) on the microstructure and mechanical properties of friction stir spot welding Al−Li alloy were discussed.

## 2. Materials and Methods

The base material used in this study was cold−rolled 2198 Al−Li alloy (thickness: 1.8 mm), and the chemical composition was obtained in Table 1. The tested state underwent T8 conditions (i.e., solution treatment, quenching, pre−deformation under tension, and artificial peak aging at 175 °C). The tensile strength of 2198 Al−Li alloy after T8 heat treatment was 510 MPa, the yield strength was 490 MPa, and the elongation is 15.5%.

For FSpW, the 2198−T8 Al−Li alloy rolling sheet was processed under a composite tool of a 9 mm shoulder and 5 mm pins in diameter, and the plunge depth of the shoulder was controlled at ~0.2 mm; the rotating speeds of the tool for FSpW were 50 rpm, 1000 rpm, and 1800 rpm (counterclockwise), with a dwelling time of 2 s. The circular diameter of the FSpW method sample was 9 mm, as shown in Figure 1.

The specimen was prepared by etching in Keller’s Reagent (5 mL HF; 3 mL HCl; 5 mL HNO_3_; 190 mL H₂O) for 15 s. The microstructure was observed using a Leica DFC−295 optical microscope. The grain size of each specimen was measured using Image Pro−Plus software. The precipitates were subsequently observed on a JEM 2100F TEM (transmission electron microscope), using the region cut from the center of each specimen. Quantitative XRD measurements were performed with a D/max2200 PC X−ray diffractometer equipped with a scanning step size of 0.02° and a scanning speed of 3 °/min. Vickers hardness testing was performed on the machined surface of the joint, using an HXZ−1000 Hardness tester with a load of 200 g and a dwelling time of 15 s. A tensile test was performed on an Instron−8801 tensile tester at room temperature under an initial strain rate of 6.7 × 10^−3^ s^−1^. The dimension distance of the tensile specimens was 3 mm, the width was 1 mm (Figure 1), and the tensile properties of the specimens were tested three times. The fractography of the tensile specimen was observed using CS−3400 SEM device.

## 3. Results

### 3.1. Microstructure

Figure 2 shows the optical micrograph and the grain sizes of the as−received specimens from the FSpW joint of the 2098−T8 Al−Li alloy. It can be observed from the macro−graph that the FSpW joint exhibited a circular zone. Its size is the same as the corresponding mixing tool and exhibited a smooth surface. Figure 2a displays a process of the specimen using a composite tool at a rotating speed of 500 rpm with a grain size of 9.1 μm. It is evident that Figure 2b showed the progress of grain refinement on the specimen, and the statistical results presented a single peak distribution of the average grain size of 9.8 μm. Figure 2c exhibits the image of the as−received specimen using the rotating speed of 1800 rpm. It can be observed from the pattern that the grain refinement process happens, the average grain size is 12.5 μm. Results show that the original “pancake” grains in the 2198−T8 Al−Li alloy can be transformed into the fine equiaxed grain and grow with the increase of rotation speed (500~1800 rpm).

Some references indicated that in the friction stir welding process, different rotation speeds will affect the temperature field distribution of Al alloys. The maximum temperature stated in that research was 513 °C at a rotational speed of 800 rpm [20]. The selection of different FSpW processing parameters has a significant impact on the temperature distribution. The maximum temperature of the FSpW Al alloy will increase due to the rise in rotation speed, but the general temperature usually does not exceed 550 °C, which is lower than the solidus temperature of Al alloys, but higher than the recrystallization temperature. In this study, after FSpW treatment, the grain structures of the base material were also uncrystallized, and a fine equiaxed crystal structure was formed into the weld stirred zone. Lower rotation speed (500 rpm) will lead to insufficient fluidity and incomplete recrystallization of some grains in Al−Li alloys. A higher rotation rate (1800 rpm) leads to full recrystallization. Appropriate rotation speed (1000 rpm) can make the grain size distribution of Al−Li alloys even.

Figure 3 shows XRD results from the FSpW Al−Li alloys at different speeds and the parent alloys. The dislocation density ρ can be assessed and calculated by the values of full width at half maximum (FWHM). Compared with the strain−free Al powder XRD curve, the XRD curved peak of Figure 3 was significantly wider. The peak broadening is usually caused by grain refinement and/or the formation of dislocations. The relationship between dislocation density, grain size, and peak broadening can be obtained specifically via the Williamson–Hall method [21].
(1)$$\frac{\beta \cos\theta }{\lambda }=\frac{0.9}{D}+\frac{2\varepsilon \sin\theta }{\lambda }$$
(2)$$\rho =14.4\varepsilon^{2}/b^{2}\;$$
where *β* is the peak width of half maximum in rad, *λ* is the wavelength of the X−ray beam (0.1542 nm for Cu Kα), *D* is the crystallite size, *ε* is the lattice strain, *θ* is the Bragg angle, *ρ* is the dislocation density, and the b is the Burgers vector.

It is reported that the effect of grain size of the peak broadening had been significant while the grain size was finer than 100 nm. In this study, the grains are much coarser than 100 nm, so the influence of grain size on peak broadening can be neglected; only the influence of dislocation needs to be taken into account. The dislocation density ρ was found via Equations (1) and (2) to be 2.3 × 10^14^, 2.1 × 10^14^, and 1.8 × 10^14^ m^−2^ for 500 rpm, 1000 rpm, and 1800 rpm, respectively. Compared to the original 2198−T8 Al−Li alloy (10.1 × 10^14^ m^−2^), it can be concluded that the dislocation density of Al−Li alloys processed via the FSpW method is lower than that of the base material. With the rising of the rotation speed, the dislocation density decreased gradually.

The effect of FSpW with different rotational speeds on the strengthening phase in the basic 2198−T8 Al−Li alloy was further studied. As shown in Figure 4, the selected area electron diffraction (SAED) patterns of the original 2198−T8 Al−Li alloy were identified using an electron beam parallel to the [$\bar{1}$12] Al axis and its corresponding schematic models are shown in Figure 4a. In the 2198−T8 Al−Li alloy, five types of phases (i.e., T1, *θ*′, β′, δ′, and S′ phases) precipitated on the matrix, according to Figure 4a. The relatively bright diffraction spots were from the Al matrix. The 1/3 [02$\bar{2}$] and 1/3 [31$\bar{1}$] diffraction spots were indexed as T1 phases (indicated by yellow circles) precipitated on the {111} Al plane of the Al matrix [22,23]. Another set of diffraction spots [02$\bar{2}$] and 1/2 [$\bar{2}$00] were indexed as super−lattice of L12−type precipitates (indicated by blue circles).

It was reported [21,24] that there were two types of strengthening phase in 2198−T8 having L12 structure: δ′ (Al_3_Li) and β′ (Al_3_Zr). In addition, diffraction spots of S′ (indicated by red circles) and *θ*′ (indicated by green circles) phases were also observed. Figure 4a displayed the BF (bright field) image of the specimen. It is evident that needle−like T1, S′, and *θ*′ phases, spherical δ′ (~10 nm) and β′ (~35 nm) phases were precipitated in the original alloy. In the DF (dark field) image (Figure 4b), a large amount of T1 phase with a length of around 50.7 nm was found, which was in good agreement with the precipitation behavior of 2198 alloys reported by Zhang et al. The secondary phase strengthening caused by these precipitates is the main reason for the increase in strength of the 2198 Al−Li alloy after T8−heat treatment.

Figure 5 shows TEM images of the FSpW 2198−T8 Al−Li alloy at different rotational speeds. According to Figure 5a, the SAED pattern of the FSpW 2198−T8 Al−Li alloy paralleled the [$\bar{1}$12] Al axis, diffraction spots of the Al matrix could be observed, and the diffraction spots from the strengthening phase were disappearing. It may be indicated that the majority of the reinforcing phases had dissolved into the Al matrix, including β′ (Al_3_Zr), θ′ (Al_2_Cu), and T1 (Al_2_CuLi). However, several β′ (Al_3_ Zr) particles were observed in the Al matrix with some dislocations in the corresponding BF images (Figure 5b,c). The diffraction intensity from the β′ (Al_3_Zr) phase is too weak to be reflected in the diffraction pattern. The local heating produced during the FSpW changed the precipitation behavior. However, the β′ (Al_3_Zr) phase can resist even high temperatures above 500 °C [16], which is much higher than the maximum temperature reached during FSPW (about 320 °C [21]). Therefore, the precipitates β′ (Al_3_Zr) are retained in the processed region after FSpW. As can be seen from the BF images, all the strengthened phases dissolved back into the original alloy. In addition, considerable dislocations were observed within the grains due to the severe shear deformation during FSpW, which is consistent with the XRD measurements.

### 3.2. Hardness Distribution

The effects of tool rotation speed during FSpW on the mechanical properties of 2198−T8 Al−Li alloy were obtained. Figure 6 shows the micro−hardness distribution of FSpW 2198−T8 Al−Li alloy at different rotational speeds, which showed a “W” shape. The micro−hardness in the weld stirred zone of 500 rpm ranges leveled off at around 100 HV_0.2_ ~ 105 HV_0.2_. The micro−hardness in the weld stirred zone of 1000 rpm varies within the range of 110 HV_0.2_ to 120 HV_0.2_.

The micro−hardness in the weld stirred zone of 1800 rpm speed ranges between 105 HV_0.2_ and 125 HV_0.2_. Compared with the micro−hardness of the 2198−T8 Al−Li alloy (180 HV_0.2_), the weld stirred zones of FSpW 2198−T8 Al−Li alloy gradually declined. The micro−hardness of 1800 rpm was relatively higher and varied from a small scale. The micro−hardness at a rotational speed of 1000 rpm is maintained at a relatively high level. It only fluctuates within a narrow range.

The decrease in the hardness of the welded joints was thought to be caused by grain recrystallization and dissolution of precipitates due to the heat generated by friction during the FSpW process. Further, the high temperature in the welded joints caused the dissolution of multiple precipitated phases into the Al matrix, which destroys the second−phase strengthening of the material and is the main reason for the reduction in strength. In addition, the heat generated by friction causes an incomplete or complete recrystallization of the grains, although the contribution of fine grain strengthening to the material is not very significant, so the difference in hardness between the three types of welded joints was not significant. 

### 3.3. Tensile Properties

The tensile properties of the specimens at different rotating speeds were further studied. Figure 7 shows the tensile properties of the 2198−T8 Al−Li alloys processed via the FSpW method.

It was indicated from Figure 7 that when the tool rotation speed of FSpW 2198−T8 Al−Li alloy was relatively slow (500 rpm) or fast (1800 rpm) the tensile strength and elongation of the weld nugget would be lower. When the rotational speed was moderate (1000 rpm), the tensile strength and elongation of the weld stirred zone would be the highest. Compared to the original 2198−T8 Al−Li alloy, after FSpW, the tensile strength and yield strength for the studied specimen exhibited a great decreasing trend, and the elongation went up. The workpiece of 1000 rpm rotation speed has the highest tensile strength, yield strength, and elongation, which were 396 MPa, 260 MPa, and 32.2%, respectively.

## 4. Discussion

Figure 8 shows the SEM images of the fracture surfaces of the ordinary 2198−T8 Al−Li alloy. As can be seen from Figure 8a, the fracture section was reduced and the necking phenomenon occurred in the tensile process, with the evidence of secondary cracks and deep dimples along the fracture surfaces. Figure 8b displays an enlarged picture of the dimple. The dimple was equiaxed in which there were broken particle phases. The results show that the fracture mechanism of the basic 2198−T8 Al−Li alloy is a quasi−cleavage fracture and the fracture mode is a ductile–brittle mixed fracture.

The SEM images of the fracture surfaces of the FSpW 2198−T8 Al−Li alloy at different rotating speeds are shown in Figure 9. The fracture section was reduced and the necking phenomenon occurred in the tensile process, as in Figure 9a,c,e. A large number of shallow dimples appeared on the surface of the specimen (Figure 9b,d,f). The fracture mechanism of the FSpW 2198−T8 Al−Li alloy specimen was micro−porous aggregation toughness fracture and the fracture mode was a ductile fracture.

Appropriate processing parameters of FSpW improved performances of plastic flow property and produced microstructure with fine grain. As a result, the mechanical properties of the as−received alloy were improved. In the FSpW process, the main affecting parameters of heat input are rotational speed and processing duration. Yang et al. [25] studied the influences of rotation speed and welding time on the 7075−T6 alloy and stated that the lower rotational speed and the shorter processing time in the 7075−T6 alloy were beneficial to the increase of mechanical property and the formation of fine microstructure. Mazzaferro et al. found the same results when using FSpW to process phase transformation−induced plastic steel (e.g., TRIP steel). In this study, slower or faster rotational speed will reduce the mechanical properties of FSpW 2198−T8 Al−Li alloy. Moderate rotation speed caused the FSpW 2198−T8 Al−Li alloy from the fine equiaxed crystal structure and uniform distribution to obtain high mechanical properties in the weld stirred zone.

## 5. Conclusions

In the present study, the influence of tool rotational speed on the FSpW joint morphology of the 2198−T8 Al−Li alloy and its mechanical properties was investigated. The following conclusions can be drawn:(1)As the rotational speed of the tool in FSpW increases, the heat input also gradually increases. This causes the average grain size of FSpW joints to gradually increase from 9.8 μm to 12.5 μm, while the dislocation density gradually decreases from 2.3 × 10^14^ m^−2^ to 1.8 × 10^14^ m^−2^.(2)Compared to the base material of 2198−T8, the joints of FSpW have reduced mechanical properties due to the effect of welding heat that causes both S’ and θ strengthening phases of the alloy to dissolve into the matrix. The fracture mode changes from mixed ductile–brittle fracture to ductile fracture.(3)After the FSpW process at a different speed, the difference in mechanical properties, separately, is due to dislocation density, grain size, and grain distribution. The mechanical properties of FSpW 2198−T8 Al−Li alloy decreased when the rotational speed was relatively slow or fast, and the mechanical properties of the studied FSpW joint were higher when the rotation speed was moderate, due to the uniformly distributed fine equiaxed crystals formed in the microstructure of the weld nugget region.

## Figures and Tables

**Figure 1 materials-16-01807-f001:**
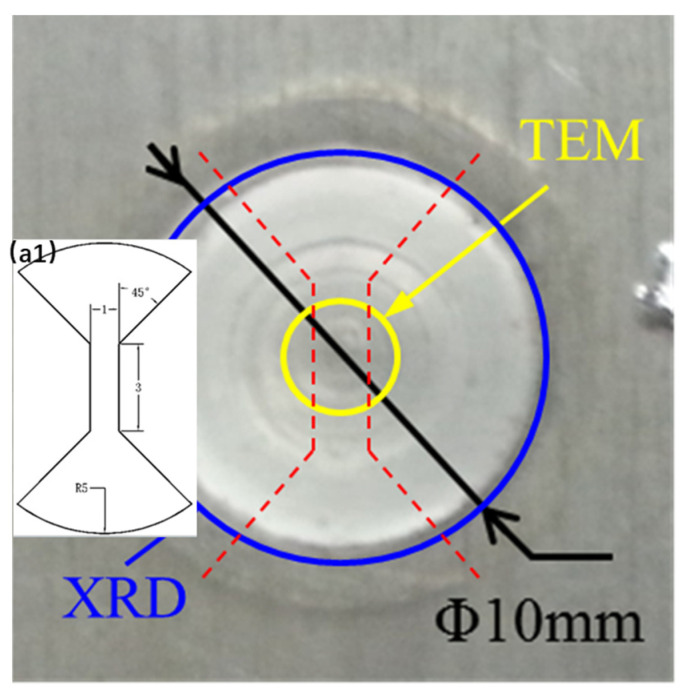
The sample via FSpW method showing dimension and location of specimens; (a1) Drawing of tensile specimen.

**Figure 2 materials-16-01807-f002:**
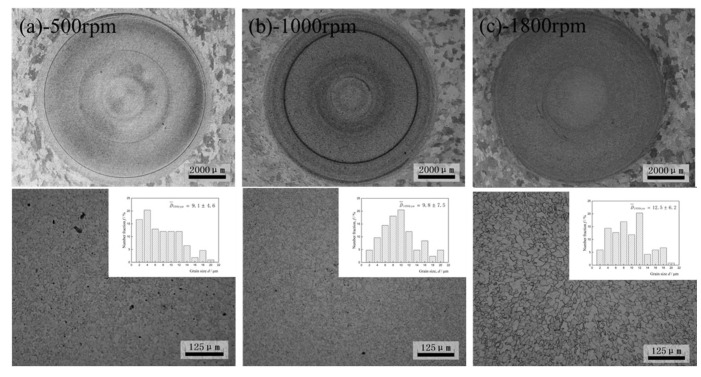
Microstructures of FSpW 2198−T8 Al−Li alloy (bottom) and Grain size distribution (top), for different rotational speeds: (**a**) 500 rpm; (**b**) 1000 rpm; (**c**) 1800 rpm.

**Figure 3 materials-16-01807-f003:**
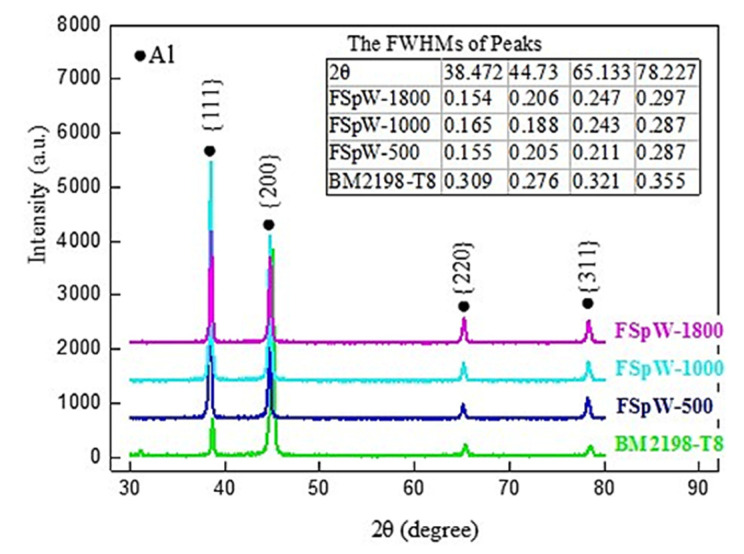
XRD curves of FSpW 2198−T8 Al−Li alloy for rotational speeds of 500, 1000, and 1800 rpm and the unstrained 2198−T8 powder.

**Figure 4 materials-16-01807-f004:**
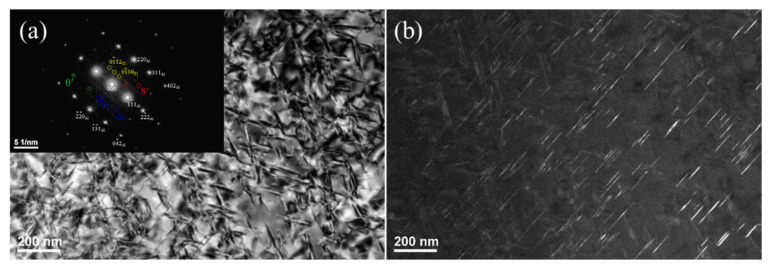
TEM images of 2198−T8 Al−Li alloy: (**a**) BF and SEAD pattern; (**b**) DF.

**Figure 5 materials-16-01807-f005:**
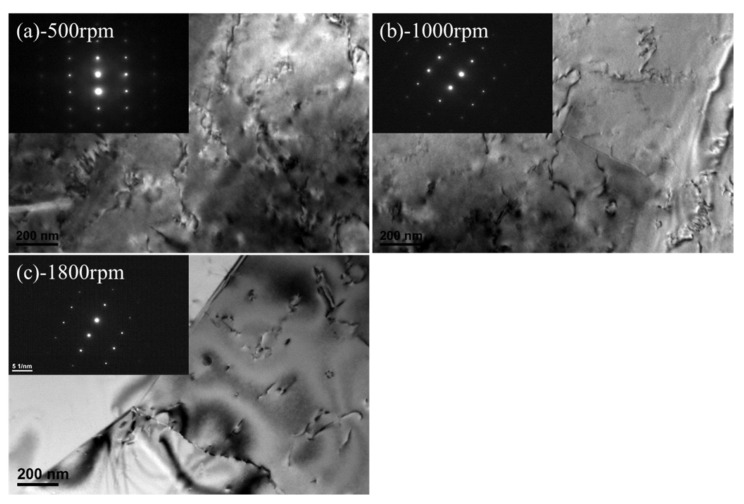
TEM images of FSpW 2198−T8 alloy at different rotational speeds: (**a**) 500 rpm; (**b**) 1000 rpm; (**c**) 1800 rpm.

**Figure 6 materials-16-01807-f006:**
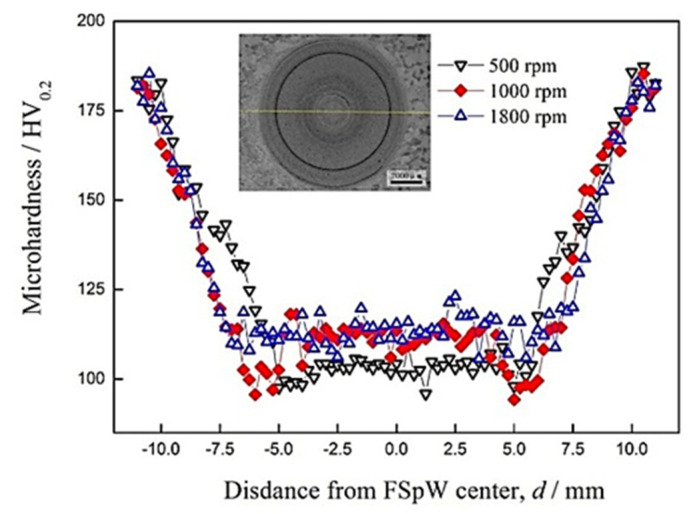
Hardness distribution of FSpW 2198−T8 Al−Li alloy at different rotational speeds.

**Figure 7 materials-16-01807-f007:**
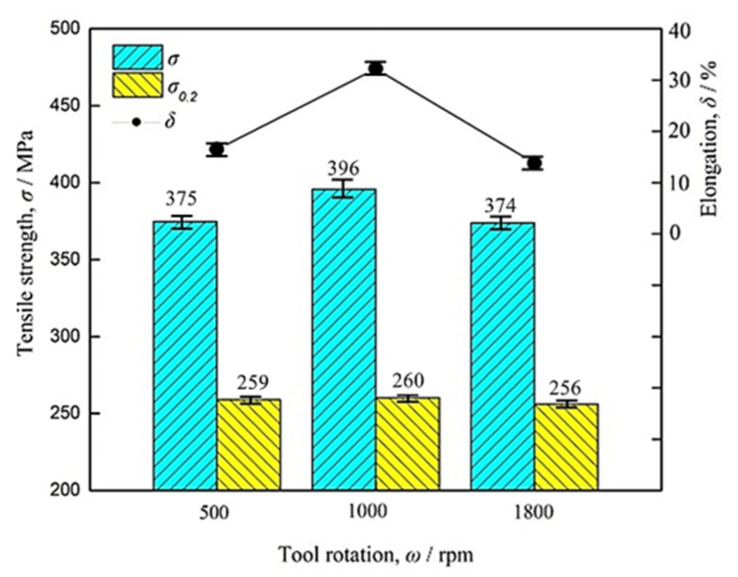
Tensile properties of 2198−T8 Al−Li alloy processed via the FSpW method at different rotating speeds.

**Figure 8 materials-16-01807-f008:**
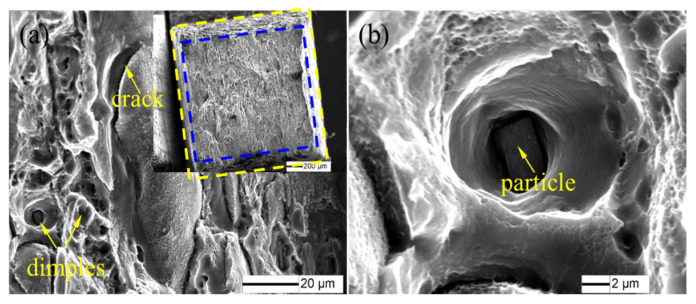
(**a**) Fractography of 2198−T8 Al−Li alloy; (**b**) local magnification.

**Figure 9 materials-16-01807-f009:**
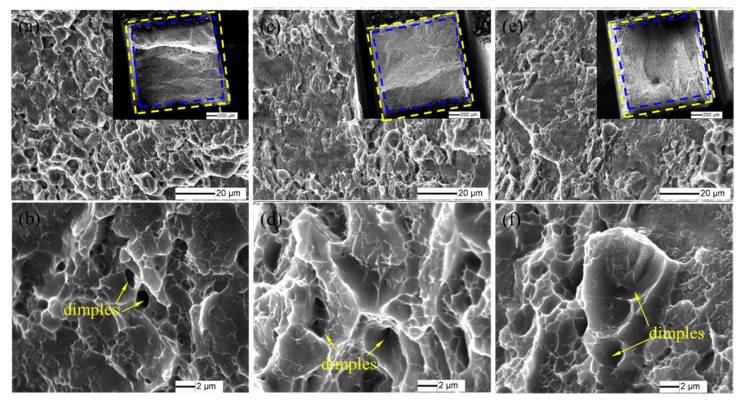
Fractography of FSpW 2198−T8 alloy at different rotational speeds: (**a**,**b**) 500 rpm; (**c**,**d**) 1000 rpm; (**e**,**f**) 1800 rpm.

**Table 1 materials-16-01807-t001:** Chemical composition of 2198−T8 Al−Li alloys (wt.%).

Alloy	Li	Cu	Mg	Zn	Zr	Mn	Ag	Al
2198	0.98	3.29	0.36	0.34	0.16	0.05	0.34	Bal.

## Data Availability

No data was generated from this work.

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
