# Peer review of "Effects of Rotational Speed on the Microstructure and Mechanical Properties of 2198-T8 Al-Li Alloy Processed by Friction Spot Welding"

_materials, 2023, doi:10.3390/ma16051807_

Round 1

Reviewer 1 Report

This research investigates the effect of rotational speed on the microstructure and mechanical properties of AL alloy processed by friction stir welding. Different parts of the article need major corrections before publishing.

The abstract should be written more attractively. Most of it contains research methods. Almost half of the abstract is devoted to generalities and research methods. The novelty of the article should be clearly added to the abstract. Use quantitative results in the abstract.

The introduction is written very superficially and briefly and needs fundamental reforms. Also, the number of references used is very small. It is suggested to use the following resources to improve it “An Exhaustive Evaluation of Fracture Toughness, Microstructure, and Mechanical Characteristics of Friction Stir Welded Al6061 Alloy and Parameter Model Fitting Using Response Surface Methodology”, “Investigation of welding crack in micro laser welded NiTiNb shape memory alloy and Ti6Al4V alloy dissimilar metals joints”, “Microstructure and mechanical properties of dissimilar NiTi/Ti6Al4V joints via back-heating assisted friction stir welding”, “Production of Al/Mg-Li composite by the accumulative roll bonding process” and “Microstructure and mechanical properties of ultrasonic spot welding TiNi/Ti6Al4V dissimilar materials using pure Al coating”.

Check the mechanical properties mentioned in the research method section (lines 68 and 69). These numbers can be unreasonable.

The number of repetitions of mechanical test samples should be mentioned. The geometry of tensile samples and standards used should also be added.

The conclusion is written very briefly. It is necessary to mention the outstanding results of the work.

Add the standard deviation and error bar to the mechanical properties data.

Most of the results sections are just reports of experimental data. More discussion and analysis of results should be done.

Reviewer 2 Report

The manuscript submitted for review requires serious revision. The authors hurried, poorly proofread the manuscript. The obtained results could be described in more detail.

1 Lots of typos for such a small manuscript:

1.1 Why AA2198-T8 is always named differently – 2198-T8 Al-Li Alloy; Al-Li 2198-T8 alloy; Al-Li 2198-T8; 2198-T8 alloy.

1.2 Why is the base metal of AA2198-T8 always named differently – unstrained Al powder BM 2198-T8; basic 2198-T8; 2198-T8 Al-Li Alloy (line 169); ordinary 2198-T8 alloy. Abidance with standard terminology is very important in technical sciences.

2 Typos on the lines: 33, 71, 95, 124.

3 What is meant by the phrase ‘used to fill keyholes’?

4 Not specified in the experimental part:

Sheet thickness

Or it was one sheet and FSpW simulated according to the principle “bead-on-bead test”.

Force and speed of sinking of pin not specified.

5 Microhardness load 200 gr. too high for this alloy, so there is no difference in results.

6 What does the abbreviation mean FWHM.

7 The explanation of the results presented in figures 4, 5, 6 can be expanded.

8 The results presented on figure 7 are within the margin of error. For a given alloy, heating due to friction leads to softening, regardless of the rotation speed. What the authors do not discuss.

9 Line 211, in my opinion, figure 9 represents SEM images, not Optical images.

10 The first point of conclusions must be removed.

Round 2

Reviewer 1 Report

Accept in present form

Reviewer 2 Report

-